# Persistent Moderate-to-Weak Mediterranean Diet Adherence and Low Scoring for Plant-Based Foods across Several Southern European Countries: Are We Overlooking the Mediterranean Diet Recommendations?

**DOI:** 10.3390/nu13051432

**Published:** 2021-04-23

**Authors:** Stefano Quarta, Marika Massaro, Mihail Chervenkov, Teodora Ivanova, Dessislava Dimitrova, Rui Jorge, Vanda Andrade, Elena Philippou, Constantinos Zisimou, Viktorija Maksimova, Katarina Smilkov, Darinka Gjorgieva Ackova, Lence Miloseva, Tatjana Ruskovska, Georgia Eirini Deligiannidou, Christos A. Kontogiorgis, Julio Sánchez-Meca, Paula Pinto, María-Teresa García-Conesa

**Affiliations:** 1Laboratory of Biochemistry and Molecular Biology, Department of Biological and Environmental Sciences and Technologies, University of Salento, 73100 Lecce, Italy; stefano.quarta3@unisalento.it; 2National Research Council (CNR), Institute of Clinical Physiology, 73100 Lecce, Italy; marika@ifc.cnr.it; 3Faculty of Veterinary Medicine, University of Forestry, 1797 Sofia, Bulgaria; vdmchervenkov@abv.bg; 4Slow Food in Bulgaria, 9 Pierre De Geytre St. bl. 3, 1113 Sofia, Bulgaria; tai@bio.bas.bg (T.I.); dessidim3010@gmail.com (D.D.); 5Department of Plant and Fungal Diversity and Resources, Institute of Biodiversity and Ecosystem Research, Bulgarian Academy of Sciences, 1113 Sofia, Bulgaria; 6Instituto Politécnico de Santarém, Escola Superior Agraria, 2001-904 Santarém, Portugal; rui.jorge@esa.ipsantarem.pt (R.J.); vanda.andrade@esa.ipsantarem.pt (V.A.); 7Life Quality Research Centre (CIEQV), IPSantarém/IPLeiria, 2040-413 Rio Maior, Portugal; 8Centro de Investigação Interdisciplinar Egas Moniz (CiiEM), Instituto Universitário Egas Moniz, 2829-511 Monte de Caparica, Portugal; 9Department of Life and Health Sciences, School of Sciences and Engineering, University of Nicosia, Nicosia 1700, Cyprus; philippou.e@unic.ac.cy (E.P.); zissimouconstantinos@gmail.com (C.Z.); 10Department of Nutritional Sciences, King’s College London, London SE1 9NH, UK; 11Faculty of Medical Sciences, University Goce Delcev, str. Krste Misirkov, No. 10-A, POB 201, 2000 Stip, North Macedonia; viktorija.maksimova@ugd.edu.mk (V.M.); katarina.smilkov@ugd.edu.mk (K.S.); darinka.gorgieva@ugd.edu.mk (D.G.A.); lence.miloseva@ugd.edu.mk (L.M.); tatjana.ruskovska@ugd.edu.mk (T.R.); 12Laboratory of Hygiene and Environmental Protection, School of Medicine, Democritus University of Thrace, Dragana, 68100 Alexandroupolis, Greece; edeligia@med.duth.gr (G.E.D.); ckontogi@med.duth.gr (C.A.K.); 13Department of Basic Psychology & Methodology, Faculty of Psychology, University of Murcia, 30100 Murcia, Spain; jsmeca@um.es; 14Research Group on Quality, Safety and Bioactivity of Plant Foods, Centro de Edafología y Biología Aplicada del Segura-Consejo Superior de Investigaciones Científicas (CEBAS-CSIC), Campus de Espinardo, P.O. Box 164, 30100 Murcia, Spain

**Keywords:** Mediterranean diet, MeDiWeB questionnaire, sex, age, body mass index, disease status, diet adherence, dietary habits, food choices, 14-MEDAS

## Abstract

The Mediterranean diet (MD) has been sponsored worldwide as a healthy and sustainable diet. Our aim was to update and compare MD adherence and food choices across several Southern European countries: Spain (SP), Portugal (PT), Italy (IT), Greece (GR), and Cyprus (CY) (MED, Mediterranean), and Bulgaria (BG) and the Republic of North Macedonia (NMK) (non-MED, non-Mediterranean). Participants (*N* = 3145, ≥18 y) completed a survey (MeDiWeB) with sociodemographic, anthropometric, and food questions (14-item Mediterranean Diet Adherence Screener, 14-MEDAS). The MED and non-MED populations showed moderate (7.08 ± 1.96) and weak (5.58 ± 1.82) MD adherence, respectively, with significant yet small differences across countries (SP > PT > GR > IT > CY > BG > NMK, *p*-value < 0.001). The MED participants scored higher than the non-MED ones for most of the Mediterranean-typical foods, with the greatest differences found for olive oil (OO) and white meat preference. In most countries, ≥70% of the participants reported quantities of red meat, butter, sweet drinks, and desserts below the recommended cutoff points, whereas <50% achieved the targets for plant-based foods, OO, fish, and wine. Being a woman and increasing age were associated with superior adherence (*p*-value < 0.001), but differences were rather small. Our results suggest that the campaigns carried out to support and reinforce the MD and to promote plant-based foods have limited success across Southern Europe, and that more hard-hitting strategies are needed.

## 1. Introduction

The Mediterranean diet (MD) has been designated as an intangible cultural heritage with a particular lifestyle and eating habits around the Mediterranean Basin. The MD dietary pattern is characterized by the preferred use of olive oil (OO; rich in monounsaturated fatty acids); a high intake of plant-based foods (fruits, vegetables, legumes, nuts); a preference for white lean meat; and a moderate consumption of fish, whole-grain cereals, and red wine. This is additionally balanced with a reduced intake of sugar-sweetened drinks, red and processed meats, whole-fat dairy products, and industrial desserts [1]. The rise in obesity and related chronic cardio-metabolic disorders continues to be a major public health problem worldwide, reflecting the failure of the current dietary models and supporting the urgent need to pursue and attain healthier lifestyle and dietary patterns [2]. In the past years, a large amount of evidence has been gathered regarding the association between the MD and, especially, the consumption of plant-based foods, with significant benefits against obesity and associated disorders [3,4]. However, despite the recognized benefits associated with the MD, populations living around the Mediterranean Sea have progressively abandoned many of their traditional dietary habits, adopting instead a more Westernized dietary pattern. This diet is characterized by an increased consumption of red and processed meat (rich in saturated fats), refined cereals, and beverages containing high levels of sugars, all being associated with increased disease risk [5]. Importantly, the adoption of healthier eating habits must be also supported by adherence to more “sustainable models” in order to safeguard the biodiversity of ecosystems, decrease global warming, and guarantee worldwide adequate dietary intakes and the survival of the succeeding generations. According to the Food and Agriculture Organization (FAO), sustainable diets should be nutritionally adequate; safe; healthy; and, at the same time, optimize both natural and human resources [6]. The MD, largely based on agricultural products, has a smaller negative impact on the environment and requires fewer natural resources (e.g., water) [7,8] than the production of meat [9], thus representing a good model of “sustainable diet” [10]. The reduction of the prevalence of diet-related chronic diseases and the increase of the environmental sustainability of dietary patterns represent two of the major 21st century food policy challenges [11].

A range of nutritional policies and guidelines including the promotion of the MD have been disseminated across Europe and worldwide to fight against the health and environmental consequences caused by the unhealthier Western dietary pattern. Yet, reinforcing the MD pattern in Mediterranean countries and/or introducing this diet in other regions of the world remains a difficult task [12]. To help achieving effective results, the development and validation of dietary and nutritional indicators that assess the population status through national and international representative population-based surveys are needed to ensure adequate monitoring activities by surveillance systems [13]. Along these lines, the 14-MEDAS score system constitutes a validated and simple dietary assessment tool of the MD adherence [14], which together with similar modified scores have been widely implemented in different Mediterranean countries, i.e., Spain [15,16], Italy [17,18], Greece [19,20], Cyprus [21], Portugal [22], Malta [23], Morocco [24,25], Israel [26], or Turkey [27]. Overall, the results of these studies have all indicated a moderate-to-low MD adherence in the adult population across the Mediterranean basin. In addition, the relationship between a range of sociodemographic (e.g., age, sex, and BMI) and lifestyle factors and MD adherence has been explored in many of those studies [15,16,17,20,23,24,25,27,28,29,30,31]. The association between MD adherence and some cardio-metabolic disorders has also been indicated [21,32]. Nonetheless, the results are highly variable and remain inconclusive.

The main aim of this cross-sectional survey was to update and compare the dietary habits across an adult sample population from several countries of the Southern European region using a common questionnaire and scoring system (14-MEDAS). For this purpose, we (1) estimated the degree of MD adherence in several Mediterranean countries (MED): Spain (SP), Portugal (PT), Italy (IT), Greece (GR), and Cyprus (CY), as well as in two countries of the neighboring Balkan region, Bulgaria (BG) and the Republic of North Macedonia (NMK) (non-Mediterranean countries, non-MED); (2) analyzed the differences between the countries regarding the preference and consumption of every specific food item included in the questionnaire; and (3) further investigated the potential association between sex, age, BMI, and disease status with the MD adherence and food choices. The results are presented and discussed in the context of previously reported similar data to determine whether there has been any improvement of the dietary habits in this region.

## 2. Materials and Methods

### 2.1. Study Ethics and Recruitment

Data for the present analysis were collected and analyzed under the frame of the MeDiWeB (Mediterranean Diet and Wellbeing) consortium constituted by several research institutions from 7 countries in the Southern European region. The project was designed to assess and compare the food habits and MD adherence across various MED and non-MED countries, as well as their relationship with an array of lifestyle and behavioral characteristics related to subjective well-being (SWB) [14,22]. The protocol was approved by the Ethics Committee of each partner research institution and complies with European Regulation on Data Protection [33]. A structured questionnaire was prepared in the official language of each country and constructed in Google Forms, being disseminated through institutional mailing lists, social media, personal contacts, and word-of-mouth communication for the collection of data. The questionnaire was confidential and filled anonymously online. Data were collected between April 2019 and mid-March 2020 (before the COVID-19 lockdown).

The initial data comprised an overall sample of 3350 adults of both sexes recruited in 5 MED countries, namely, SP, PT, GR, CY, IT, and 2 Balkan countries, NMK and BG (non-MED countries). Participants were eliminated from the study due to (i) lack of consent, (ii) age <18 y, (iii) duplicates, and/or (iv) participants whose nationality differed from the country in which they were living in. A total of 3145 adults (age ≥ 18 years) distributed across the different countries were finally included in the study. Inadequate responses (lack of food item information) were additionally eliminated for the analyses of the 14-MEDAS score (Appendix A).

### 2.2. MeDiWeB Questionnaire

The MeDiWeB questionnaire was designed according to the Organization for Economic Co-operation and Development (OECD) recommendations [34] and has been already fully described and published [22]. In brief, the questionnaire includes items on sociodemographic data, SWB, health status, lifestyle, and dietary choices. In the current study, we report the results of the analyses of the following subset of variables for each of the participant countries and for the MED group and non-MED groups: sociodemographic data (age, sex), health-related information (self-reported diagnosed pathology, weight (kg), and height (m), used to calculate body mass index (BMI, kg/m^2^)), and food choices. Participants were also classified into (1) age groups (early (18–24 y), middle (25–44 y), and late (45–64 y) adults and senior (≥65 y)), (2) BMI categories (underweight (<18.5kg/m^2^), normal weight (18.5–24.9 kg/m^2^), overweight (25.0–29.9 kg/m^2^), and obesity (≥30 kg/m^2^)) as defined by the WHO [35], and (3) disease status (no declared pathology, 1 pathology, multi-pathology). The nature of the pathologies was classified using the International Statistical Classification of Diseases and Related Health Problems [36].

The food questions included the 14 items that constitute the 14-MEDAS screener adapted from PREDIMED and that was used to calculate the 14-MEDAS score as previously described [14]. The results were ranked to estimate the adherence to the MD as follows: weak adherence, ≤5; moderate adherence, 6–9; good adherence ≥10 [37]. The MeDiWeB survey also included several questions about daily consumption of milk and derived products; preference for low-fat dairy products and whole cereals; daily intake of water, caffeinated drinks, and herbal teas; and the total number of meals per day.

### 2.3. Statistical Analyses

The normality of the continuous variables in the different population samples was assessed by graphical inspection (frequency histograms and Q-Q plots), as well as by the Kolmogorov–Smirnov and Shapiro–Wilk tests. In most cases, the variables were significantly deviated from a normal distribution and thus we herein report the results of comparative non-parametric tests. Absolute frequencies and percentages were used to represent ordinal or nominal variables (sex, age range, BMI range, disease status, food choices), whereas the scale variables (age, BMI, 14-MEDAS score) were presented as the median and interquartile range (IQR). The mean and SD were also included to simplify comparison with previously published data.

The Mann–Whitney test (M-W) was used to assess the differences in the 14-MEDAS score between sexes, whereas the Kruskal–Wallis (K-W) test was applied for comparisons across age ranges, BMI categories, and disease status subgroups. Chi-squared tests (Chi-S) were used to assess the statistical association between nominal and ordinal variables. Partial correlation between continuous variables was calculated using the Spearman correlation coefficient (*ρ*) adjusted in each case for the corresponding confounding variables (sex, age, BMI, disease). The general linear model was used to compare the 14-MEDAS score through countries while controlling for the effects of sex, age, BMI, and disease. We additionally quantified the effect size by means of the standardized mean difference [38]. The results were interpreted following Cohen’s guidelines (0.2 small, 0.5 medium, and 0.8 large, in absolute value [39]). Fisher’s exact test was applied to assess whether the differences between the frequency of observations were significant between the MED and non-MED group of countries. The strength of the association between these 2 groups was additionally estimated by the Phi and Cramer’s *V* association coefficients (following Cohen’s guidelines, values were classified as 0.1 small, 0.3 medium, 0.5 large, in absolute value [39]). All statistical tests were based on two-sided tests (bilateral significance) and had a significance level of 5% (α = 0.05). All analyses were conducted using the Statistical Package for the Social Sciences (SPSS, version 26.0; IBM Corp., Armonk, NY, USA).

## 3. Results

### 3.1. Characteristics of the Participants

The characteristics of the sample population in each country and in the MED and non-MED entire groups are shown in Table 1. Overall, this population was constituted by a higher proportion of female than male participants (ratio ≈7:3). There was a slightly higher percentage of women in the non-MED countries than in the MED countries (71.4% vs. 67.2%, *p*-value < 0.02). Within the MED countries, the highest proportion of women was found in GR (77.7%), followed by CY and PT (72%), IT (62.4%), and SP (56.8%). In the non-MED participants, differences in the percentage of women between BG (69.1%) and NMK (74.1%) did not reach statistical significance.

The participants had an average age of ≈38 y, with the bulk of the sample population (93%) classified as middle and late adults (25 to 64 y), and only a small percentage (≈3%) were senior participants (≥65 y). Comparative mean age analyses across countries showed small but significant differences (*p*-value < 0.001), with SP and BG having the oldest population samples (45.7 ± 12.8 y and 43.0 ± 12.8 y, respectively), while the participants from GR and NMK were the youngest (34.8 ± 8.9 y and 29.2 ± 10.5 y, respectively). We did not detect significant differences for BMI between MED and non-MED countries, with a common average value of 24.6 kg/m^2^ and the highest proportion of participants classified in the normal weight category (58.4% and 56.7%, for the MED and non-MED groups, respectively). In all the countries, the sample population was constituted by more than 30% of overweight and obesity participants and less than 5% of underweight participants. At the country level, we found some small but significant differences with the highest BMI mean values and the highest percentage of overweight and obesity participants in GR, BG, and SP (47.4%, 42.9%, and 40.8%, respectively), while NMK showed the lowest rate (32.9%).

The participants were asked to indicate whether they had any diagnosed pathology and, if so, to describe which type of disease it was. The distribution of the responses is also included in Table 1. Regardless of the nationality, most of the participants (65–85%) were considered healthy since they did not report any pathology, and only a very small proportion of individuals (<8%) declared having more than one pathology. We found some significant differences (*p*-value < 0.001) between the MED and the non-MED groups, with ≈10% more participants in the non-MED countries that did not report any pathology as compared to the MED ones. Across the individual countries, SP exhibited the largest proportion of participants with one or more pathologies, while NMK, BG, and IT showed the highest rate of healthy individuals. Regarding the nature of the pathologies, a summary with the principal reported diseases is presented in Appendix A. The most commonly declared disorders were those related to the endocrine, nutritional, or metabolic diseases (≈20–40%), followed by diseases of the circulatory system (≈15–28%). In a lower proportion (but above 10%), the participants also reported having diseases of the digestive system (IT, PT, and BG), respiratory system (PT and GR), and musculoskeletal and connective tissue system (CY and GR).

### 3.2. Analysis of the 14-MEDAS Score across Countries

The 14-MEDAS scores, their distribution into the three main categories of MD adherence, and the comparative analyses between sexes, age ranges, BMI categories, disease status, and across countries are included in Appendix A (MED countries) and Appendix A (non-MED countries). Results are also presented and summarized in Figure 1. Within the MED group, 68.3% of the participants were classified into the moderate category of MD adherence, with median score values between 6.0 and 8.0 in all the subgroups analyzed. On average, only 11.0% of the MED participants were classified into the high MD adherence group (score ≥ 10.0). The participants of the non-MED group exhibited slightly lower scores than the MED group, with median values between 6.0 (moderate) and 5.0 (low) and with the highest proportion (51.3%) classified into the low category of MD adherence. Differences between the MED and non-MED sample populations reached high statistical significance (*p*-values < 0.001 or <0.01) in all the subgroups analyzed. Differences across the countries were small but significant and can be discerned looking at the mean and SD values. The population from SP exhibited the highest score (7.90 ± 1.72) followed by PT (7.38 ± 2.10), GR (7.13 ± 2.03), and IT (6.80 ± 1.54), whereas the CY population had the lowest value (6.12 ± 1.99) of the MED countries. BG and NMK had significantly lower scores (5.80 ± 1.80 and 5.30 ± 1.80, respectively). The significant differences between countries and ranking order were sustained in most of the subgroups examined.

Regarding the differences between sexes, women exhibited a slightly higher mean score than men in all the countries (standardized mean difference between sexes of ≈0.2, categorized as small in the Cohen scale [39]). Sex differences reached statistical significance in the entire MED and non-MED groups as well as in the PT, SP, and BG sample population (Figure 1a). The 14-MEDAS score was generally found to be slightly higher with increasing age (Figure 1b,c; Spearman adjusted correlation coefficients between 0.1 and 0.3, small effect size in the Cohen guideline [39]). Differences across age ranges were significant both in the whole MED group and in the non-MED group, as well as for most individual countries (except for BG). In the case of the sample populations from GR and BG, the score increased only until the 45–64 y age range and then it decreased in participants 65 y old or older; however, these results may be affected by the low number of participants in the senior age range. With respect to the classification of the 14-MEDAS score into the main BMI categories, we detected a minor downwards trend of the 14-MEDAS score with increasing BMI in only some of the MED countries (Figure 1d), although the changes were rather small (Spearman adjusted correlation ≈ −0.1) and only reached statistical significance in the sample populations from CY, GR, and PT. In the non-MED countries, we did not detect a significant trend between the 14-MEDAS score and BMI, although the score also decreased with increasing BMI category in the case of the sample population from BG (Figure 1d,e). We were not able to detect significant differences between the 14-MEDAS score across the three categories of the disease status (Figure 1f,g).

### 3.3. Comparison across Countries of the 14-Food Items Included in the 14-MEDAS Score

We next analyzed and compared the choices for each of the food items (included in the 14-MEDAS score) across all countries as well as between the entire MED and non-MED groups. A summary with a comprehensive contrast between the MED group and the non-MED group is displayed in Figure 2. Individual results are listed in Table 2 and can also be seen as radar graphs in Appendix A.

Overall, the MED group had a higher percentage of participants who scored 1 (within Mediterranean diet recommendations) than the non-MED group for most of the food choices excluding sweet drinks, which was similar in both groups of countries, and wine consumption, which was slightly higher in the non-MED participants, especially in BG.

Nevertheless, most differences between the two groups were categorized as small, except for the preference for OO as the main culinary fat (94.7% in the MED group vs. 38.0% in the non-MED group; 0.62, large difference based on the Phi and Cramer’s V association coefficients; Table 2). Within the MED countries, the best score for OO preference was detected in SP and the lowest in CY, yet the estimation of the quantity of OO consumed in all the countries was well below the MED diet recommendations (less than 30% of the participants declared to consume ≥4 tbsp/day). With regards to the consumption of fats (butter and cream), more than 65% of the participants in each country declared eating <1 portion per day, with the lowest consumption detected in IT and SP and the highest in BG and PT.

In both the MED and non-MED countries, more than 95% of the participants consumed <1 portion red meat/day. This result was in reasonable agreement with a high preference for white meat (≈70% or higher) in the MED countries. The preference for white meat was slightly lower (less than 60%) in the participants of the non-MED countries. The consumption of fish was very low in all the countries, with PT and SP exhibiting the highest percentage of participants consuming ≥3 portions of fish per week (35.0% and 19.4%, respectively).

The consumption of vegetables, fruits, legumes, and nuts were in general very low, with most countries exhibiting around 50% or less of the participants with intakes within the MD recommendations (Table 2). Values were particularly low for the IT and NMK sample populations, wherein the proportion of participants consuming the recommended portions of these four items were 41%, 8.7%, 14.3%, and 10.3%, and 21.3%, 8.1%, 14.5%, and 18.4%, respectively. The consumption of ≥2 meals with “sofrito” per week in the MED countries was about 60 to 75%, with the exception of CY, where only ≈30% of people were within this recommendation. Desserts achieved percentages above 70% within recommendations for most countries, except for NMK, where only 58% of the sample population indicated consuming <3 portions/week.

As already stated, there were no significant differences between the average MED and non-MED groups with respect to the consumption of sweet drinks and, although there were some small significant differences between the individual countries, they all had more than 65% of participants consuming <1 sweet beverage per day. On the other hand, the proportion of participants in each country consuming the recommended amount of wine (7 to 14 glasses/week) was very low, mostly ≤5%, except for BG, which reached 15% of the sample population (Table 2).

We additionally explored the differences in food choices between sexes, age ranges, BMI categories, and disease status (Appendix A). Overall, there were no outstanding differences between the analyzed subgroups, with most results being non-significant (particularly in the non-MED group) or with very small differences between them (<0.2 strength of association or effect size by Phi and Cramer’s V association coefficients). Regarding sex (Appendix A), women in general scored higher than men for most food items, but the most noticeable differences were found for white meat preference, which was higher in women than in men both in the MED and the non-MED groups. The consumption of vegetables was also significantly higher in women in the MED countries, whereas in the non-MED countries, women scored slightly better for the consumption of red meat and sofrito. In general, and within the MED group of countries, most food choices were slightly improved with age (Appendix A), except for the consumption of fish, butter, or cream, as well as the preference for white meat. The improvement with age was less clearly seen in the non-MED group, although we still detected some small improvement in the consumption of desserts, sweet drinks, vegetables, and wine, principally up to the 45–64 y age range. Differences in individual food choices between BMI categories and disease status (Appendix A, respectively) were mostly not significant or very small, with some food items slightly worsened across categories, whereas some others were improved (i.e., small decrease of the white meat preference with BMI increase in the MED group or small improvement in the scoring of sweet drinks with BMI in the non-MED group).

### 3.4. Additional Food-Related Questions Included in the MeDiWeB Questionnaire

We further analyzed and compared the responses to the remaining food questions included in the MeDiWeB survey but not used in the calculation of the 14-MEDAS score. The results are presented in Table 3 and in Appendix A which, in general, show small differences across the countries. Overall, ≥60% of the participants from each country indicated consuming “one or two milk and derived products per day”, but the highest percentages were seen in SP (85.2%), IT (73.5%), and GR (70.2%). There was a significantly higher preference for low-fat dairy products in the MED group than in the non-MED group (64.2% vs. 35.4%, *p*-value < 0.001). The preference for whole grain varied across the countries from ≈50% (BG and SP) to nearly 80% (CY). The percentage of participants reporting to consume 1.0 to 2.0 L of water per day and one to three cups of caffeinated drinks was ≈50% to 60%, and values were similar in both groups of countries. On the other hand, the percentage of participants consuming one to three cups of herbal teas was ≤35% in most countries, except for BG, which had the highest value (45.3%).

We finally compared the number of meals per day in each country and found that, on average, the participants from the MED countries had 1 meal more per day than the non-MED countries (4.0 ± 1.0 and 3.1 ± 0.9, respectively) (Table 3).

## 4. Discussion

In recent years, populations living in Southern Europe and around the Mediterranean basin have experienced a general shift from their traditional dietary habits towards more Westernized patterns. Efforts to reverse this situation and its associated increased disease risk have been implemented throughout the past years with different strategies, including the promotion of the MD and, in particular, of the healthier and more sustainable plant-based foods [41]. Yet the progress of these efforts had not been simultaneously assessed in a cross-national study using a common method to measure the degree of adherence to the MD. The current investigation compares for the first time the MD adherence and food choices in a large sample of adults from a group of Southern European MED countries (SP, PT, IT, GR, CY) and non-MED countries of the Balkan region (BG, NMK) applying the same methodology.

Data were collected using a web-based survey that included the 14-MEDAS scoring system previously validated in the same countries [14]. In that former preliminary validation study (*N* < 100 participants per country), the estimated 14-MEDAS score was able to discern between the MED participants with a moderate and higher score than the Balkan participants, which showed a weak adherence. The results from the present study (*N* ≈ 300–500 participants per country) confirmed the difference between a moderate score for the MED participants and a lower adherence for BG and NMK. The sample population from SP was also established to have the highest score in both studies. Even so, the specific MD adherence classification of the participant countries (SP > PT > GR > IT > CY > BG > NMK) was slightly altered as compared with that attained in the previous validation study (SP > IT > PT > CY ~ GR > NMK > BG) [14], showing that for countries with very little differences between them, a precise ranking of their MD adherence would benefit from the implementation of cross-national studies applying common protocols and scoring systems. Larger sample populations will still be required to further substantiate the small differences across individual countries detected in our study.

Regarding the MED participant countries, analogous results with scores in the midrange of the scale were previously reported in SP [15,16,28,30,37,42,43], PT [22,44,45], IT [17,18,46,47,48,49], GR [50,51], and CY [52]. Other Southern European as well as some non-European MED countries [23,27,53] have also been attributed a moderate value of MD adherence. There is, however, limited information regarding MD adherence in countries from the Balkan region. According to the Mediterranean adequacy index (MAI), countries such as Albania or BG have also experienced, in the past decades, a considerable drift from the Mediterranean dietary habits [54], and a cross-sectional study carried out in Croatia (*N* = 10,001 participants) reported a Mediterranean Diet Serving Score (MDSS) of ≈45% of the maximum score [55]. The PREDIMED 14-item food frequency questionnaire was only recently applied in a study conducted in Albania (*N* = 209 participants), showing that the majority of the participants had also a moderate MD adherence [56]. Our study is the first to apply the 14-MEDAS scoring system to estimate MD adherence in BG and NMK, confirming that the participants from these two countries had a moderate-to-low MD adherence. All the above data evidence that, with independence of the scoring system employed, countries in Southern Europe including those classified as typically MED countries as well as countries of the Balkan region, which traditionally had food patterns with common characteristics with the MD, have all consistently reported to have a dietary pattern substantially different from the traditional MD. This difference is positioned ≈6–7 points below the highest 14-MEDAS score and does not appear to have improved during recent years.

To determine which specific food items were responsible for the observed moderate MD adherence values and the differences between the investigated countries, we analyzed the percentage of participants surpassing the Mediterranean cutoff criteria depicted in the 14-MEDAS scoring system. We found evidence of both positive and negative features in the MED populations taking part in this study. On the favorable side, and in good agreement with earlier studies also conducted in the general adult population from SP [15,16,28,30,37,42], PT [44,45], IT [17], or GR [19], our results supported a clear preference for OO against butter, with OO displaying more than 90% of subjects within the MD recommendations, and between ≈60% and 95% consuming less than one portion of butter or cream per day. Nonetheless, the target quantity for OO (≥4 tbsp/day) was reached by ≤30% of the participants. Previous results related to the quantity of OO intake [15,16,17,19,26,28,30,37,42,44,50] show, in general, a high variability suggesting some difficulty in the estimation and reporting of the actual daily intake of this product. On the other hand, the consumption of sofrito (prepared with OO) was also reasonably high (>60% of participants within recommendations except for CY) and support a frequent consumption of OO as the main cooking oil in these MED countries [15,17,19,30,37,42,44]. The results of our survey also pointed towards a reduction of red meat consumption, with more than 95% of the MED participants indicating achieving the MD recommendations. Similar results were previously reported in GR [19], but more variable and generally lower percentages were reported in other studies conducted in SP [15,16,28,37,42], IT [17], and PT [44], suggesting a potential improvement in these countries with regards to the consumption of red meat. In support of this, our results showed a high proportion (≈70–85%) of participants preferring white meat, also in good agreement with some data previously reported in SP [37,42], IT [17], and PT [44]. In addition, our data pointed to a reduced consumption of sugar, with ≈65–80% of participants achieving the recommended targets for the intake of sweet beverages and desserts. These results are similarly supported by previous data in SP [15,16,28,37,43], IT [17], PT [44], and GR [19]. With regard to plant-based foods, our analyses indicated a low intake of fruits, vegetables, legumes, and nuts in the MED countries, with percentages of participants achieving the targets well below 50%. These results are consistent with those reported in previous studies conducted in SP, IT, PT, and GR [15,16,17,19,28,37,42,44]. Of note, our study also showed a very low percentage (≤5%) of compliance with MD recommendations for wine drinking. These results were even lower than previous reported scores for this item [15,17,28,37,42,44], suggestive of a reduction in wine intake. Fish consumption was also found to be, in general, low (<20–30%), and even lower than previously reported in SP and IT [15,17,28,37,42,44].

Overall, the percentage of participants scoring within the MD recommendations for most food items was lower in the two Balkan countries than in the MED countries. The largest difference was found for OO, with only ≈40% of the Balkan participants preferring this fat source. These results are in reasonable agreement with the latest data for OO supply quantity (kg/person y) as estimated by the Food and Agriculture Organization [57] in those same countries. On the other hand, a high proportion of the non-MED participants indicated complying with the recommendations for reduced intake of red meat and of sweet beverages. These results are to some extent in contrast with an earlier subtle but steady raising trend in the consumption of pork and processed meat products [58] and a visible increase of soft drink consumption for BG [59,60]. Our results for the consumption of plant-based foods in BG and NMK also indicated a substantial deviation from the MD recommended targets. Although there is limited published information about food choices in these countries, previous studies had indicated an important need to increase the consumption of dietary fiber in BG and NMK [61,62], already suggesting a low consumption of plant-based foods. A significant drop of plant food intake in BG, especially of fresh fruit, was seen during the post-communist years [63], although some more recent research has shown that about 50% percent of university students habitually consumed fruits and vegetables [64], suggesting a certain improvement in younger generations. Conversely, another study conducted in adolescents from NMK has shown that less than 20% of them were consumers of the recommended daily servings of fruit and vegetables [65]. The MD recommendations for nuts, legumes, and fish were also reached by a very low percentage of participants in BG and NMK, which may be partially attributed to the high prices of some of those products as well as a lack of tradition, for example, of fish eating [66]. On the other hand, the results of the consumption of wine in BG exhibited the best response as compared with the rest of investigated countries, with 15% of respondents achieving the 7 to 14 glasses per week. This higher value may be, however, associated with customary uses taking place during the wintertime in this country [67]. Recently, and more in agreement with a low percentage of participants achieving wine drinking targets, wine consumption in BG was reported to have declined [68].

Taking into account all the evidence reported here, we were able to infer that in the general adult population from the Southern European countries examined, there appears to be a consistent and persistent good intake of OO and white meat, especially in the MED countries, concomitant with a reduced intake of butter and red meat as well as a reduced intake of high-sugar products (beverages, desserts). However, there is still a rather low consumption of fruits, vegetables, legumes, nuts, fish, and wine, underlying the 6–7 points difference between the moderate MD adherence score estimated in these countries and the highest scores achievable as described in the 14-MEDAS analysis.

To fully comprehend the differences in MD adherence between different populations, researchers have long investigated the influence of a range of sociodemographic and lifestyle factors. In particular, differences across sexes, age groups, or BMI categories have been examined in various MED countries with inconsistent outcomes. Some studies have reported no clear differences in MD adherence between sexes in SP and IT [15,16,17,37], whereas other studies have indicated a higher adherence in women than in men in SP, IT, and PT [17,45,48,55]. Regarding age, some studies conducted in SP have reported no significant association between age and MD adherence [16,37]. In IT, a lower adherence was reported in younger people [17], whereas other studies carried out in SP [15,69] and Croatia [55] have indicated that elderly people have a better adherence to MD than young people. The results of previous analyses looking at the association between MD adherence and BMI in various MED countries are also variable. In GR [70], Morocco [25], and IT [71], an inverse association between BMI and MD adherence was reported. In SP, however, some studies have shown no significant differences across BMI categories [16], whereas others have indicated either a higher MD adherence associated with a higher BMI [15] or an inverse association, with subjects with higher BMI reporting a lower MD adherence [37]. The results of our study further support (i) a higher MD adherence in women than in men, and (ii) a positive association between age and MD adherence with significant increases from younger to older participants. These results were generally seen both in the MED and the non-MED participants. It should be noted that participants ≥ 65 y were in much lower numbers than the younger groups (partially due to online questionnaires potentially being more accessible for young people) and thus further studies with increasing numbers of elderly participants are required to corroborate the association between age and MD adherence. Our results also point to a downwards trend of the MD adherence with increasing BMI that was only significant in some of the MED participant countries. Our study also supports the small size of the differences in MD adherence between sexes, age groups, and BMI categories (0.1 to 0.3 in the Cohen scale), which can be attributed to small differences at the level of most of the individual food items rather than larger differences in a few specific food items.

We additionally explored some other traditionally featured Mediterranean food items not included in the 14-MEDAS, such as whole-grain cereals and low-fat milk products. The intake of whole-grain cereals, which contributes to the overall energy intake [72] and has been associated with a lower risk of chronic diseases and a healthier aging [73,74,75], has decreased in the last years partly due to the increased consumption of more palatable ultra-processed grain products and of alternative products such as gluten-free foods [76]. The results in our study show, however, some preference for whole grains (as indicated by ≥50% of the responders), which was also slightly but significantly higher in the MED countries than in the non-MED participants. On the other hand, and although the association between the consumption of dairy products in general and the risk of chronic diseases is not yet clear [77,78,79], a recent study has confirmed a protective role of the low-fat dairy products against the development of metabolic syndrome in a Mediterranean population of the PREDIMED cohort [80]. Our results show a small but significantly higher daily intake of dairy products in the MED than in the non-MED participants, but, importantly, there was a marked difference in the preference for the consumption of low-fat milk, yogurt, and cheese, which was much lower in the Balkan countries, and especially in BG, probably associated with a long tradition of the consumption of full-fat milk products such as yogurt or cheese in this country [81]. Our data support the relevance and interest in evaluating these food items within the context of the studies about MD adherence.

Chronic dehydration has been associated with a worsening of cardio-metabolic disease risk [82,83,84,85], and a positive association has been observed between MD adherence and water intake in a Spanish elderly cohort of the PREDIMED study [86]. In our study, we asked the participants about the consumption of water and other beverages such as caffeinated drinks and herbal teas. The importance of adequate total water intake is invoked by the major national and international health authorities including the European Food Safety Agency (EFSA), which recommends between 2.0 and 2.5 L/d for adult females and males including drinking water, water from other beverages, and water present in food [87]. Within the MD, water (1.5–2.0 L/d) is recommended as the main source of hydration to preserve body water balance and maintain an active lifestyle [40,88]. In our MED and non-MED cohorts, the percentage of responders that consumed 1.0 to 2.0 L/day was ≈50% to 60%. The consumption of caffeinated drinks reached similar levels, whereas the herbal tea consumption was the lowest, with values below 40%, except in BG, certainly leaving space for some improvement. In addition, we show a slightly lower meal frequency in the non-MED countries (three per day) against the MED-countries (four per day). Whether the meal frequency per day has a role in cardio-metabolic health remains controversial, but some studies suggest that a higher number of meals per day may have a positive regulatory effect in some metabolic parameters such as cholesterol levels [89]. The degree to which the difference in the daily number of meals may contribute to the MD adherence remains to be established. The quantity and quality of those meals is also an important issue. It is plausible that if more frequent snacks favor, for example, a higher intake of fruit and raw vegetables, this may contribute to improve MD adherence [90]. Further research into these issues would support the development of better recommendations on sustainable and healthy meal planning.

In an attempt to improve their populations’ health and lifestyle, all the countries included in this study have developed and applied national level dietary guidelines that follow the MD recommendations for most of the food items examined in this study, especially for OO, fruits, vegetables, legumes, fish, water, sugar-sweetened beverages, desserts, and red and processed meats. NMK, however, has not yet included OO and wine in their recommendations, whereas nuts are specifically recommended in the guidelines from GR, PT, SP, NMK, and BG, and a moderate consumption of wine in the guidelines from CY, GR, PT, SP, and BG [91]. Despite the existence of these guidelines, the results of our analysis and of previous similar studies indicate that, overall, a high proportion of the population from the MED and non-MED countries in Southern Europe continue to report unhealthy dietary habits, especially low intake of plant-based foods, disregarding some of the main MD recommendations. It remains essential, thus, to establish how to best promote behavior changes towards a higher MD diet adherence, both in MED and non-MED populations. As recently explained by Fernández-Lázaro et al. [92], the several points improvement of the MD adherence in participants of the PREDIMED study was achieved after one-year of hard work of a highly trained multidisciplinary team combined with a novel model of personalized recommendations, motivation, and direct communication with the participants. The translation of this success to the general population is not an easy task, and a different combination of schemes have been repeatedly proposed: (1) impulse stronger and country-adapted food mandatory policies and guidelines (e.g., increase the promotion and availability of traditional, locally produced, and sustainable plant-based foods); (2) reinforce and improve communication and information about the MD as well as the means to get the messages across the population (social media, public events, local retailers, nutritional services); and (3) increase the population knowledge and understanding of the health benefits but also and, importantly, of the sustainability of the MD through specific education programs with a especial focus on younger generations [93]. In this context, the development of national and international networks merging the experience and knowledge of researchers, medical and nutritional professionals, food producers and retailers, policymakers, and consumers would be a positive asset to further achieve our goals [94]. Along these lines, the recent revision and updating of the MD pyramid including the sustainability aspect constitutes an excellent example of a joint effort of a multidisciplinary team from different areas working towards the promotion of the MD [88].

To the best of our knowledge, this is the first time a web survey (MeDiWeB questionnaire) has been concurrently distributed across the general free adult sample population from several countries in the Southern Europe, including MED and non-MED representatives, with the purpose of evaluating and comparing the adherence to the MD using the same index, i.e., the 14-MEDAS. This is also the first time this tool has been used in two representatives of the Balkan countries, BG and NMK. We acknowledge that the responders may not fully represent the general population in these countries, and that self-reported data might be subjected to some recall bias, leading to some over or underestimation of the consumption of the different food items. However, since the 14-MEDAS score system is a well-accepted and widely employed method and our procedure achieved a reasonable response rate, we believe that these data provide reliable information about the dietary pattern and main differences between the participant countries. Although we have included in our study the analysis of the association between the 14-MEDAS and specific factors (sex, age, and BMI), we also acknowledge that additional lifestyle and socioeconomic factors can influence these results. Further, the cross-sectional design of the current study does not allow for evaluating food pattern variations over the year (seasonality) or to determine any cause–effect relationship. Future studies are warranted to investigate all these other aspects.

## 5. Conclusions

In conclusion, this web-based study conducted across several MED and non-MED countries in Southern Europe corroborates a persistent deviation from the MD adherence by a substantial proportion of the adult population in this region and reinforces the need to impulse efforts towards the improvement of the dietary habits, more specifically, to increase the current low consumption of plant-based foods. The MeDiWeB consortium involved in this research is constituted by researchers from different institutions across the Mediterranean and Balkan regions aiming at monitoring the progress of the MD implementation in all these countries and contributing to the promotion of a healthier and more environmentally sustainable world.

## Figures and Tables

**Figure 1 nutrients-13-01432-f001:**
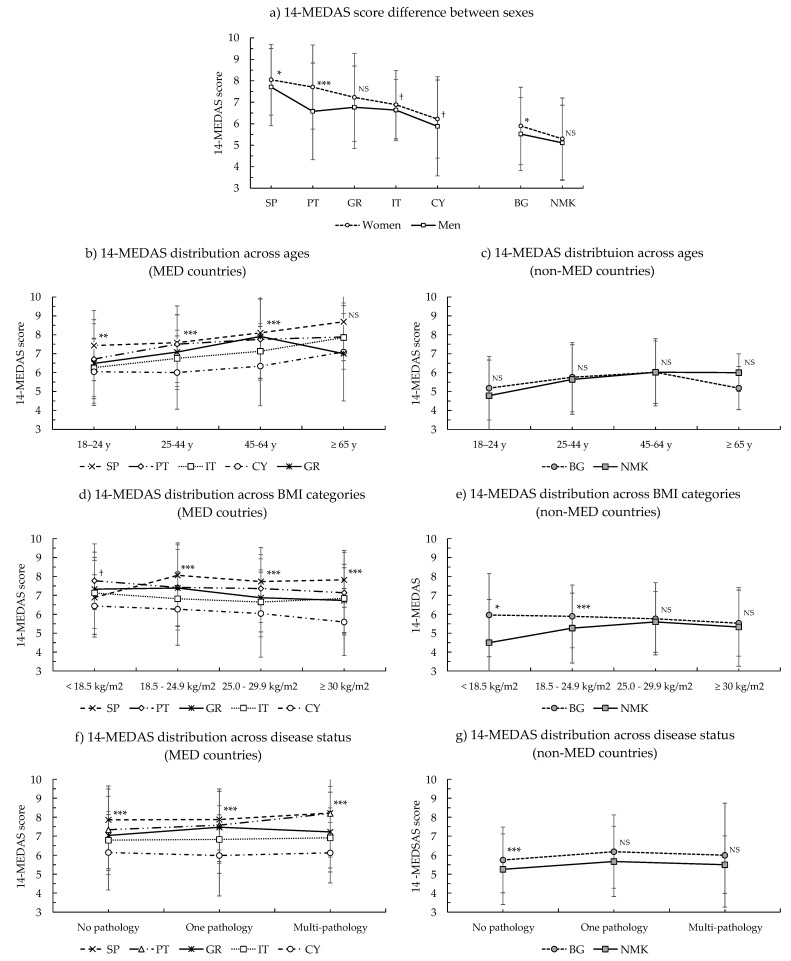
14-MEDAS score comparative analyses: (**a**) differences between sexes and countries, (**b**) distribution across age ranges in the MED countries, (**c**) distribution across age ranges in the non-MED countries, (**d**) distribution across BMI categories in the MED countries, (**e**) distribution across BMI categories in the non-MED countries, (**f**) distribution across disease status subgroups in the MED countries, (**g**) distribution across disease status subgroups in the non-MED countries. PT: Portugal; SP: Spain; IT: Italy; GR: Greece; CY: Cyprus; BG: Bulgaria; NMK: Republic of North Macedonia; MED: Mediterranean countries; Non-MED: non-Mediterranean countries. NS: not significant; † *p*-value < 0.1; * *p*-value < 0.05; ** *p*-value < 0.01; *** *p*-value < 0.001 significant differences between sexes in (**a**) and between countries in (**b**–**g**).

**Figure 2 nutrients-13-01432-f002:**
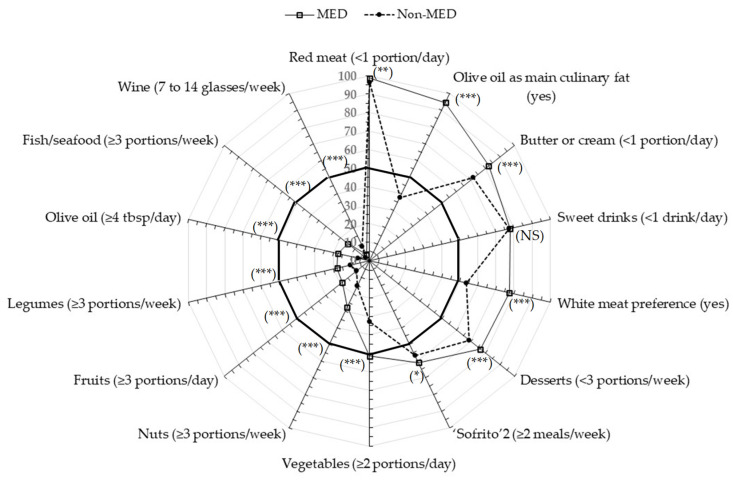
Rank-ordered radar plot of the food items included in the 14-MEDAS score comparing the choices between the MED group of countries and the non-MED group. Data represents the percentage of participants in each group scoring 1 for each question (within the Mediterranean diet adherence recommendation). Dark intermediate circle represents the 50% point. NS: not significant; * *p*-value < 0.05; ** *p*-value < 0.01, *** *p*-value < 0.001.

**Table 1 nutrients-13-01432-t001:** Characteristics and distribution of the sample population.

	SP	IT	PT ^1^	GR	CY	MED	BG	NMK	Non-MED
Total *N* (%)	486 (21.9)	505 (22.8)	488 (22.0)	296 (13.4)	442 (20.0)	2217	492 (53.0)	436 (47.0)	928
Sex									
Women; *N* (%)	276 (56.8)	315 (62.4)	349 (72.0)	230 (77.7)	317 (72.0)	1487 (67.2)	340 (69.1)	323 (74.1)	663 (71.4)
Men; *N* (%)	210 (43.2)	190 (37.6)	136 (28.0)	66 (22.3)	123 (28.0)	725 (32.8)	152 (30.9)	113 (25.9)	265 (28.6)
Age (years)									
Median (IQR)	49.0 (18.0)	36.0 (25)	34.0 (23.8)	33.0 (12.0)	37 (16)	38.0 (23.0)	42.0 (17.0)	24.0 (14.0)	35.0 (22.0)
Mean ± SD	45.7 ± 12.8	38.1 ± 14.2	36.5 ± 13.6	34.8 ± 8.9	38.1 ± 12.1	39.0 ± 13.3	43.0 ± 12.8	29.2 ± 10.5	36.7 ± 13.6
(*N*)	(485)	(505)	(488)	(293)	(442)	(2213)	(492)	(434)	(926)
Range (years)	18–74	18–80	18–75	18–66	19–75	18–80	18–90	18–74	18–90
18–24 (years) *N* (%)	46 (9.5)	118 (23.4)	124 (25.4)	26 (8.9)	64 (14.5)	378 (17.1)	30 (6.1)	221 (50.9)	251 (27.1)
25–44 (years) *N* (%)	143 (29.5)	215 (42.6)	215 (44.1)	228 (77.8)	262 (59.3)	1063 (48.0)	250 (50.8)	157 (36.2)	407 (44.0)
45–64 (years) *N* (%)	281 (57.9)	150 (29.7)	141 (28.9)	38 (13.0)	104 (23.5)	714 (32.3)	186 (37.8)	53 (12.2)	239 (25.8)
≥65 (years) *N* (%)	15 (3.1)	22 (4.4)	8 (1.6)	1 (0.3)	12 (2.7)	58 (2.6)	26 (5.3)	3 (0.7)	29 (3.1)
BMI (kg/m^2^)									
Median (IQR)	24.2 (4.4)	23.5 (4.9)	23.4 (5.2)	24.6 (6.6)	23.8 (6.2)	23.8 (5.3)	24.2 (6.6)	23.3 (5.3)	23.7 (6.2)
Mean ± SD	24.7 ± 3.9	21.1 ± 4.1	24.2 ± 4.6	25.6 ± 5.5	24.6 ± 4.7	24.6 ± 4.5	25.2 ± 5.7	23.8 ± 4.1	24.6 ± 5.1
(*N*)	(481)	(501)	(474)	(293)	(440)	(2189)	(491)	(422)	(913)
^2^ Underweight (<18.5 kg/m^2^) *N* (%)	9 (1.9)	22 (4.4)	17 (3.6)	9 (3.1)	21 (4.8)	78 (3.6)	24 (4.9)	21 (5.0)	45 (4.9)
Normal weight (18.5–24.9 kg/m^2^) *N* (%)	276 (57.4)	307 (61.3)	296 (62.4)	145 (49.5)	254 (57.7)	1278 (58.4)	256 (52.2)	262 (62.1)	518 (56.7)
Overweight (25.0–29.9 kg/m^2^) *N* (%)	150 (31.2)	134 (26.7)	109 (23.0)	85 (29.0)	100 (22.7)	578 (26.4)	126 (25.6)	111 (26.3)	237 (26.0)
Obesity (≥30 kg/m^2^) *N* (%)	46 (9.6)	38 (7.6)	52 (11.0)	54 (18.4)	65 (14.8)	255 (11.6)	85 (17.3)	28 (6.6)	113 (12.3)
(Overweight + obesity) *N* (%)	196 (40.8)	172 (34.3)	161 (34.0)	139 (47.4)	165 (37.5)	833 (38.0)	211 (42.9)	139 (32.9)	350 (38.3)
Disease status									
Non declared pathology *N* (%)	317 (66.6)	423 (83.8)	342 (74.7)	220 (74.6)	336 (76.0)	1638 (75.3)	401 (83.9)	364 (86.1)	765 (84.9)
Declared one pathology *N* (%)	123 (25.8)	71 (14.1)	106 (23.1)	66 (22.4)	73 (16.5)	438 (20.1)	65 (13.6)	53 (12.5)	118 (13.1)
Declared multi-pathology *N* (%)	36 (7.6)	11 (2.2)	10 (2.2)	9 (3.1)	33 (7.5)	100 (4.6)	12 (2.5)	6 (1.4)	18 (2.0)

*N*: Sample size; ^1^: data in this column are from [22], except for age categories, and are included here for comparison with the rest of countries; ^2^: BMI categories were defined according to WHO [35]; SP: Spain; IT: Italy; PT: Portugal; GR: Greece; CY: Cyprus; MED: all Mediterranean countries together; BG: Bulgaria; NMK: Republic of North Macedonia; Non-MED: all non-Mediterranean countries together. Variation in sample size is due to missing data in the different variables.

**Table 2 nutrients-13-01432-t002:** Food choices in the sample population for each individual country and each group of countries: percentage of participants scoring 1 (within Mediterranean diet recommendations) for each question of the 14-MEDAS. SP: Spain; IT: Italy; PT; Portugal; GR: Greece; CY: Cyprus; BG: Bulgaria; NMK: Republic of North Macedonia. MED: all Mediterranean countries together; Non-MED: all non-Mediterranean countries together.

14-MEDAS Question (Score 1)	SP	IT	PT	GR	CY	BG	NMK	MED	Non-MED	*p*-Value (Chi-S) ^1^	*p*-Value (Fisher) ^2^	Effect Size ^3^
1. Olive oil as main culinary fat (yes)	98.3	97.0	96.5	96.6	84.6	43.7	31.8	94.7	38.0	<0.001	<0.001	0.62 (large)
2. Olive oil (≥4 tbsp ^4^/day)	16.5	21.4	9.4	28.3	15.2	9.2	3.1	17.4	6.4	<0.001	<0.001	0.14 (small)
3. Vegetables (≥2 portions/day)	54.3	41.0	53.1	56.8	54.9	43.9	21.3	51.5	33.1	<0.001	<0.001	0.17 (small)
4. Fruits (≥3 portions/day)	25.0	8.7	23.8	16.8	18.9	9.1	8.1	18.8	8.6	<0.001	<0.001	0.13 (small)
5. Red meat (<1 portion/day)	99.2	98.6	95.5	100.0	99.5	95.3	97.7	98.4	96.4	<0.001	0.001	0.06 (small)
6. Butter or cream (<1 portion/day)	91.9	95.0	67.8	77.1	76.8	68.9	75.0	82.3	71.8	<0.001	<0.001	0.12 (small)
7. Sweet drinks (<1 drink/day)	88.2	77.2	82.6	67.8	68.3	84.5	69.9	77.8	77.5	<0.001	NS	0.003 (NS)
8. Wine (7 to 14 glasses/week)	2.5	5.8	4.3	2.1	2.7	15.0	1.4	3.6	8.6	<0.001	<0.001	0.10 (small)
9. Legumes (≥3 portions/week)	20.2	14.3	23.0	6.9	20.4	7.0	14.5	17.8	10.6	<0.001	<0.001	0.09 (small)
10. Fish/seafood (≥3 portions/week)	19.4	6.5	35.0	2.1	5.2	3.0	1.9	14.8	2.5	<0.001	<0.001	0.18 (small)
11. Desserts (<3 portions/week)	81.6	71.7	72.5	73.5	82.9	78.5	58.0	76.5	68.8	<0.001	<0.001	0.08 (small)
12. Nuts (≥3 portions/week)	36.6	10.3	29.7	42.3	26.1	12.3	18.4	27.8	15.2	<0.001	<0.001	0.13 (small)
13. White meat preference (yes)	84.6	68.8	76.8	73.7	82.4	49.7	58.0	77.4	53.7	<0.001	<0.001	0.24 (small)
14. “Sofrito” ^5^ (≥2 meals/week)	75.5	63.4	68.0	67.7	30.3	55.4	58.1	61.1	56.7	<0.001	<0.05	0.04 (small)

^1^ Chi-squared tests (Chi-S) were used to assess differences between all individual countries; ^2^ Fisher’s exact tests were used to compare the MED and non-MED groups of countries; ^3^ Phi and Cramer’s *V* association coefficients between the MED and non-MED countries (strength of association or effect size: small 0.1, median 0.3, large 0.5); ^4^ tbsp: table spoon; ^5^ “Sofrito”: mixture of tomato, onion, and/or garlic fried in olive oil use for cooking meals. NS: not significant.

**Table 3 nutrients-13-01432-t003:** Preferences of the sample population in each country for the remaining food-related questions of the MeDiWeB questionnaire. Percentage of participants scoring the indicated quantity. SP: Spain; IT: Italy; PT; Portugal; GR: Greece; CY: Cyprus; BG: Bulgaria; NMK: Republic of North Macedonia. MED: all Mediterranean countries together; Non-MED: all non-Mediterranean countries together.

Question (Within MD Recommendations)	SP	IT	PT	GR	CY	BG	NMK	MED	Non-MED	*p*-Value (Chi-S) ^2^	*p*-Value (Fisher) ^3^	Effect Size ^4^
Milk and derived products (one or two/day) ^1^	85.2	73.5	60.8	70.2	61.3	59.1	62.4	70.4	60.7	<0.001	<0.001	0.18 (small)
Preference for low fat milk, yogurt, or cheese (yes) ^1^	68.0	66.6	46.8	62.4	77.1	31.3	39.9	64.2	35.4	<0.001	<0.001	0.26 (median)
Preference for whole grain (yes)^1^	55.3	65.2	68.7	66.5	79.2	49.5	73.2	67.3	59.6	<0.001	<0.001	0.07 (small)
Water (one to two liters/day) ^1^	59.1	47.9	59.1	49.0	61.3	53.3	67.3	57.7	56.3	<0.001	NS	0.03 (small)
Caffeinated drinks (one to three/day)	64.8	68.6	58.2	60.4	58.1	62.6	56.3	60.1	63.8	<0.001	NS	0.04 (small)
Herbal teas (one to three/day)	35.2	23.9	28.0	18.5	34.1	45.3	22.8	26.0	40.0	<0.001	<0.001	0.18 (small)
Number of meals (per day)	3.9 ± 0.9 ^5^ 4.0 (2.0) ^6^	3.7 ± 1.0 4.0 (1.0)	4.2 ± 1.0 4.0 (1.0)	4.1 ± 1.0 4.0 (2.0)	4.0 ± 1.2 4.0 (2.0)	3.2 ± 1.0 3.0 (1.0)	2.9 ± 0.9 3.0 (1.0)	4.0 ± 1.0 4.0 (2.0)	3.1 ± 0.9 3.0 (2.0)	<0.001 ^7^	<0.001 ^8^	-

^1^ Based on Cena and Calder [40]; ^2^ Chi-squared tests (Chi-S) were used to assess differences between all individual countries; ^3^ Fisher’s exact tests were used to compare the MED and non-MED groups of countries; ^4^ Phi and Cramer’s *V* association coefficients between the MED and non-MED countries (strength of association or effect size: small 0.1, median 0.3, large 0.5); ^5^ mean value ± SD; ^6^ median (interquartile range); ^7^ Kruskal–Wallis test for independent samples (comparison between all countries). ^8^ Mann–Whitney test for independent samples (comparison between MED and non-MED countries); NS: not significant.

## Data Availability

Results attained in this study are included in the manuscript and in the Appendix A. Individual data are not publicly available due to ethical restrictions.

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
