# Peer review of "Persistent Moderate-to-Weak Mediterranean Diet Adherence and Low Scoring for Plant-Based Foods across Several Southern European Countries: Are We Overlooking the Mediterranean Diet Recommendations?"

_nutrients, 2021, doi:10.3390/nu13051432_

Round 1

Reviewer 1 Report

This is a well-written and detailed paper, describing the adherence to the MD in several Mediterranean countries and 2 non-Mediterranean countries. It also investigated the data by sex, age, BMI and disease status. 

I have only a few minor comments:

Generally, be consistent about whether to include single inverted commas around the MEDAS question groups

Line 61 – olive oil in full before abbreviation

Lines 199 – 201 – clarify that the percentages relate to women

In Table 1 & other tables, highlight Med & non-Med columns, so that they stand out more

Rows 321-335 – include where to find data – Table 2

Line 343 – correct spelling of association

Supp tables S3-S6 – suggest separate columns for p-value1 and effect size2

Supp table S6 – 8,8 instead of 8.8

Lines 375 & 549 – whole grain & Table 3

Table 3 – need for ‘other beverages’ row?

Line 385 – In recent years,

Line 426 – ‘countries such as Albania and BG’

Line 441 – replace ‘the past years’ with ‘recent years’        

Is it possible to shorten the discussion, without losing relevant information?

Is it possible to include a detailed description of the 14 MEDAS questions, perhaps as a supplementary table? I don’t think there is a need to include the question number in the figures.

Reviewer 2 Report

Τhe MS is well written and presented.Do you have any data about portion size, frequency of meals. How did you interpret consumption of local type of foods?

Author Response

We thank the reviewer for taking the time and effort to go through our study and for her/his consideration of our research.

Regarding the frequency of meals, we have included in Table 3 information regarding the number of meals per day in each country. These results, although preliminary, show that there are differences between some countries and this may be an important factor that influences our diet and must be investigated further. This question was part of the MeDiWeb questionnaire, already published and made available as Supplementary material in reference 22 of the manuscript. The reviewer can also see in this questionnaire that the food-related questions were based on an indicated average portion. The questionnaire was also previously validated with FFQ questionnaires (24 h, 3 days) as reported in reference 14 of the manuscript.

In this current research, the food-related questions were all general as shown in Table 2, and we did not investigate specific local foods. Indeed, it is important to investigate further local specific foods, especially from the point of view of diet sustainability. Future studies are warranted in this direction.